# MmpS5-MmpL5 Transporters Provide *Mycobacterium smegmatis* Resistance to imidazo[1,2-*b*][1,2,4,5]tetrazines

**DOI:** 10.3390/pathogens9030166

**Published:** 2020-02-28

**Authors:** Dmitry A. Maslov, Kirill V. Shur, Aleksey A. Vatlin, Valery N. Danilenko

**Affiliations:** Laboratory of Bacterial Genetics, Vavilov Institute of General Genetics, Russian Academy of Sciences, Moscow 119333, Russia; shurkirill@gmail.com (K.V.S.); vatlin_alexey123@mail.ru (A.A.V.); valerid@vigg.ru (V.N.D.)

**Keywords:** *Mycobacterium smegmatis*, imidazo[1,2-*b*][1,2,4,5]tetrazine, MmpS5-MmpL5, efflux, drug discovery, drug resistance, tuberculosis

## Abstract

The emergence and spread of drug-resistant *Mycobacterium tuberculosis* strains (including MDR, XDR, and TDR) force scientists worldwide to search for new anti-tuberculosis drugs. We have previously reported a number of imidazo[1,2-*b*][1,2,4,5]tetrazines–putative inhibitors of mycobacterial eukaryotic-type serine-threonine protein-kinases, active against *M. tuberculosis*. Whole genomic sequences of spontaneous drug-resistant *M. smegmatis* mutants revealed four genes possibly involved in imidazo[1,2-*b*][1,2,4,5]tetrazines resistance; however, the exact mechanism of resistance remain unknown. We used different approaches (construction of targeted mutants, overexpression of the wild-type (*w.t.*) and mutant genes, and gene-expression studies) to assess the role of the previously identified mutations. We show that mutations in *MSMEG_1380* gene lead to overexpression of the *mmpS5-mmpL5* operon in *M. smegmatis*, thus providing resistance to imidazo[1,2-*b*][1,2,4,5]tetrazines by increased efflux through the MmpS5-MmpL5 system, similarly to the mechanisms of resistance described for *M. tuberculosis* and *M. abscessus*. Mycobacterial MmpS5-MmpL5 transporters should be considered as an MDR-efflux system and they should be taken into account at early stages of anti-tuberculosis drug development.

## 1. Introduction

Tuberculosis (TB), caused by *Mycobacterium tuberculosis*, is currently the leading killer among the infectious diseases caused by one infectious agent, responsible for an estimate of 1.2 million deaths in 2018 [1]. According to WHO, 1.7 billion people globally are infected with *M. tuberculosis*, and thus are at risk of developing the disease [1]. The emergence and spread of multidrug resistant TB (MDR-TB, defined as TB resistant to rifampicin and isoniazid), extensively drug-resistant TB (defined as MDR-TB with resistance to the fluoroquinolones and second-line injectables) and totally drug-resistant TB (TDR-TB) is a global threat to world-wide TB control [2,3,4,5]. Long treatment times (six months for drug-susceptible TB and up to two years for DR-TB) often lead to bad patient compliance, which is one of the causes of drug resistance development and results in worse treatment outcomes. Thus, researchers are forced to search for novel anti-TB drugs and shorter regimens [6].

Eukaryotic-type serine-threonine protein-kinases (ESTPKs) play a key role in *M. tuberculosis* life cycle regulation, controlling some of its vital aspects such as cell division and survival within host macrophages, and, therefore, they represent attractive targets for drug development [7,8]. We have previously described a number of imidazo[1,2-*b*][1,2,4,5]tetrazines with a promising antibacterial activity on *M. tuberculosis* and *M. smegmatis* [9]. Most of these compounds showed activity as potential ESTPK inhibitors in the original *M. smegmatis aphVIII+* test-system [10,11], and two of them were able to bind to the *M. tuberculosis* PknB adenine-binding pocket according to docking studies [10]. Despite the predicted activity as ESTPK-inhibitors, both the exact mechanism of action and the mechanism of resistance to these compounds are still unknown.

We were able to obtain spontaneous *M. smegmatis* mutants resistant to four imidazo[1,2-*b*][1,2,4,5]tetrazines (**3a**, **3c**, **3h** and **3n**, Figure 1), which had cross-resistance among them, suggesting a common mechanism of drug-resistance [9]. Whole-genomic sequencing and comparative genomic analysis revealed mutations in *MSMEG_0641* (binding-protein-dependent transporters inner membrane component) in 1 mutant, in *MSMEG_1601* (hypothetical protein) in seven mutants, in *MSMEG_2087* (transporter small conductance mechanosensitive ion channel (MscS) family protein) in one mutant [12], while all the mutants carried different mutations in *MSMEG_1380* (AcrR/TetR_N transcriptional regulator) – 1 nonsynonymous SNP, 2 insertions leading to a frameshift, 2 duplications (6 and 501 base pairs-long) and one deletion [13].

In this article we describe the investigation of these mutations’ role in mycobacterial drug resistance to imidazo[1,2-*b*][1,2,4,5]tetrazines by different approaches: construction of targeted mutants, overexpression of the wild-type (*w.t.*) and mutant genes, and gene-expression studies.

## 2. Results

### 2.1. Mutations in MSMEG_1380 Gene Lead to Imidazo[1,2-b][1,2,4,5]tetrazines Resistance in M. smegmatis

The list of nonsynonymous mutations found in spontaneous drug-resistant *M. smegmatis* mutants used in this study is presented in Table 1. We were able to construct targeted *M. smegmatis* mutants harboring each mutation in genes *MSMEG_0641*, *MSMEG_1601*, and *MSMEG_2087*, as well as five mutations found in *MSMEG_1380* gene using the p2NIL/pGOAL19 suicide system [14] for homologous recombination (Table 1).

We examined the minimal inhibitory concentrations (MICs) of the four compounds on the recombinant *M. smegmatis* strains and found that mutations in *MSMEG_1380* gene lead to elevated MICs as compared to the *w.t.* strain (4×MIC for the compound **3a**, at least 2×MIC for the compound **3c**, at least 2×MIC for the compound **3h,** and 4×MIC for the compound **3n**), while recombinants harboring mutations in genes *MSMEG_0641*, *MSMEG_1601*, and *MSMEG_2087* had the same MICs as the *w.t.* strain (Table 2).

Thus we have shown that only the mutations in *MSMEG_1380* are responsible for imidazo[1,2-*b*][1,2,4,5]tetrazines resistance in *M. smegmatis*.

### 2.2. W.t. MSMEG_1380 Overexpression Increases M. smegmatis Susceptibility to imidazo[1,2-b][1,2,4,5]tetrazines

In order to investigate further the role of *MSMEG_1380* in *M. smegmatis* resistance to imidazo[1,2-*b*][1,2,4,5]tetrazines, we cloned the *w.t. MSMEG_1380* gene and two of its mutant variants in the tetracycline inducible plasmid pMINDKm^-^ [15].

We used the paper-disc assay to assess the drug susceptibility of *M. smegmatis* strains to imidazo[1,2-*b*][1,2,4,5]tetrazines and found that the overexpression of the *w.t. MSMEG_1380* gene increases *M. smegmatis* susceptibility to the tested compounds, while the overexpression of its mutant variants had no effect on the phenotype (Table 3), thus suggesting that the disruption of MSMEG_1380 protein’s function leads to the drug-resistant phenotype.

### 2.3. MSMEG_1380 Represses the Expression of the mmpS5-mmpL5 Operon in M. smegmatis

*MSMEG_1380* gene lies 179 b.p. upstream the *mmpS5-mmpL5* operon (genes *MSMEG_1381-MSMEG_1382*) in the *M. smegmatis* genome and is transcribed in the opposite direction (Figure 2). The structure of this operon is conserved in different mycobacterial species: the *mmpS5-mmpL5* genes are controlled by a TetR-family transcriptional repressor, encoded by a gene located upstream the operon. Mutations in genes encoding the TetR-repressor, which lead to the upregulation of the *mmpS5-mmpL5* genes, are involved in *M. abscessus* resistance to the derivatives of thiacetazone [16], as well as in cross-resistance of *M. tuberculosis* to bedaquiline and clofazimine [17]. We have also identified a possible operator sequence in the 5′-untranslated region of *MSMEG_1381*, similar to the one described in [16]: 5′-AAGCGGATTGACCTTATCCACTT-3′.

To test the hypothesis that resistance to imidazo[1,2-*b*][1,2,4,5]tetrazines in *M. smegmatis* has a similar origin to the ones described for *M. tuberculosis* and *M. abscessus* [16,17], we analyzed the expression of *MSMEG_1380* gene and *mmpL5* (*MSMEG_1382*) genes in different conditions.

All the spontaneous *M. smegmatis* mutants had increased *mmpL5* expression (54.16–80.45 times) as compared to the *w.t. M. smegmatis mc2 155* strain (Figure 3A). The overexpression of the *w.t. MSMEG_1380* gene, cloned into the pMINDKm^-^ plasmid led to a 7.90-fold repression of the *mmpL5* gene expression (p < 0.001, Figure 3B), confirming that *MSMEG_1380* encodes the repressor of the *mmpS5-mmpL5* operon, and explaining the drug-susceptible phenotype, observed in the *MSMEG_1380* overexpressing strain. On the contrary, the expression of *MSMEG_1380* was upregulated in the mutant strains (Figure 3A), indicating that this transcriptional repressor is self-regulatory and that mutations lead to the loss of its function.

We also observed that the addition of subinhibitory concentrations of the compound **3a** upregulated the expression of *mmpL5* in a dose-dependent manner (Figure 3C).

## 3. Discussion

Deorphaning phenotypic screening hits, that is determining their mechanism of action and/or resistance, is a key part in the early-stage anti-TB drug development [18]. Here we determine the mechanism of *M. smegmatis* imidazo[1,2-*b*][1,2,4,5]tetrazines resistance based on the previously obtained whole-genome sequencing data for 12 spontaneous mutants with cross-resistance to four compounds [9,13].

The construction of targeted mutants showed that only mutations in *MSMEG_1380* are responsible for drug resistance. In *M. smegmatis*, *MSMEG_1380* encodes a TetR-family transcriptional repressor, which controls the *mmpS5-mmpL5* operon, encoding transmembrane transporters, conserved throughout mycobacterial species. Mutations occurring in *MSMEG_1380* led to the upregulation of the *mmpS5-mmpL5* operon and increased efflux of the drug-candidates from the cells, similarly to the mechanisms described for *M. tuberculosis* and *M. abscessus* [16,17]. Overexpression of the *w.t. MSMEG_1380* led to the repression of the *mmpS5-mmpL5* operon and expectedly to an increased drug susceptibility phenotype. Interestingly, we observed a dose-dependent upregulation of the *mmpS5-mmpL5* operon upon the addition of one of the compounds, which may indicate the ability of this compound to bind to the MSMEG_1380 protein, inhibiting its affinity to the operator sequence; however, this needs to be examined in vitro in future studies.

The tested compounds showed activity as ESTPK inhibitors [9,11]; however, we have not observed any mutations in ESTPK genes. One or more ESTPKs might still be the biotargets of imidazo[1,2-*b*][1,2,4,5]tetrazines but determining them by spontaneous mutagenesis might be difficult: some of the ESTPKs may fulfill the functions of others in the situation when they might be inhibited [8], and there is a possibility that more than one mutation might be required.

The primary biological role of the MmpS5-MmpL5 system consists in siderophore transport, which is crucial for *M. tuberculosis* survival under low-iron conditions within macrophages [19]. Yet, this efflux system has also shown itself to be an important factor of drug resistance: besides the mentioned efflux-mediated resistance to thiacetazone derivatives, bedaquiline and clofazimine [16,17], it has also been reported to provide *M. tuberculosis* resistance to azoles [20]. We can expect that *M. tuberculosis* strains resistant to bedaquiline and clofazimine might also be resistant to imidazo[1,2-*b*][1,2,4,5]tetrazines; however, a 36% mismatch in the amino acid sequences of the MmpL5 proteins in *M. smegmatis* and *M. tuberculosis* may affect the drug-specificity of the transporter, and this should be additionally examined in future studies. Still, the MmpS5-MmpL5-mediated resistance mechanism needs to be considered during early stages of anti-TB drug development, and convenient in silico and in vitro test-systems for rapid analysis should be developed.

## 4. Materials and Methods

### 4.1. Bacterial Strains and Growth Conditions

*M. smegmatis* strains described in this study are presented in Table 1. Middlebrook 7H9 medium (Himedia, India) supplemented with OADC (Himedia, India), 0.1% Tween-80 (v/v), and 0.4% glycerol (v/v) was used as liquid medium, while the M290 Soyabean Casein Digest Agar (Himedia, India) was used as solid medium. *Escherichia coli* DH5α was used for plasmids propagation. Cultures in liquid medium were incubated in the Multitron incubator shaker (Infors HT, Switzerland) at 37 °C and 250 rpm.

### 4.2. Targeted M. smegmatis mutants’ Construction

Targeted *M. smegmatis mc2 155* mutants were constructed by homologous recombination using the p2NIL/pGOAL19 suicide vector system [14]. Briefly, genes *MSMEG_0641, MSMEG_1380, MSMEG_1601*, and *MSMEG_2087* with adjacent 1-1,5 kb fragments were amplified from genomic DNA, isolated from respective mutants by phenol-chloroform/isoamyl alcohol extraction after enzymatic cell lysis [21], with Phusion High-Fidelity DNA Polymerase (Thermo Scientific, USA) using the following primers, picked with primer-BLAST [22]: pN_0641_f 5′- TTTTCTGCAGCCAACAACGATCCAGATGTCCGT-3′ and pN_0641_r 5′- TTTTAAGCTTCAATGGCGGCGTCTTCATTCTG-3′ for *MSMEG_0641*; pN_1380_f 5′- TTTTAAGCTTGTACTACTCGCTGGTGGCGTC-3′ and pN_1380_r 5′- TTTTGGATCCTGCTGCACGTGTTCGGTGTC-3′ for *MSMEG_1380*; pN_1601_f 5′- CCCATGACGGGCATCATCAACC-3′ and pN_1601_r 5′- TTTTTTAATTAACGACGATCAGCACGTCCACAC-3′ for *MSMEG_1601*; pN_2087_f 5′- TTTTAAGCTTCCAGAAGGTCACCAGCGATCTG-3′ and pN_2087_r. The amplified products were digested with respective restriction enzymes (Thermo Scientific, USA) and ligated in the p2NIL plasmid. The cassette from pGOAL19 was subsequently cloned in the obtained plasmids at the *Pac*I restriction site. The plasmids were electroporated in *M. smegmatis mc2 155* cells as described in [23] and plated on M290 plates supplemented with kanamycin (50 μg/mL), hygromycin (50 μg/mL), and X-Gal (50 μg/mL); blue single-crossover colonies were selected. Blue colonies were grown overnight in liquid 7H9 medium with ADC, and serial 10-fold dilutions were plated on M290 plates supplemented with X-Gal (50 μg/mL) and sucrose (2% w/v); white double-crossover colonies were selected and tested for Km susceptibility. Target genes were then Sanger-sequenced for a final confirmation of the mutation.

### 4.3. MIC Determination

MICs of the studied compounds on *M. smegmatis* were determined in liquid medium. *M. smegmatis* strains were cultured overnight in 7H9 medium, then diluted in the proportion of 1:200 in fresh medium (to approximately OD_600_ = 0.05). 196 μl of the diluted culture was poured in sterile nontreated 96-well flat-bottom culture plates (Eppendorf, Germany) and 4 μL of serial two-fold dilutions of the tested compounds in DMSO were added to the wells. The plates were incubated at 37 °C and 250 rpm for 48 h. The MIC was determined as the lowest concentration of the compound with no visible bacterial growth.

### 4.4. MSMEG_1380 Cloning, Expression and Drug-Susceptibility Testing

*MSMEG_1380* genes from respective strains were amplified by Phusion High-Fidelity DNA Polymerase (Thermo Scientific, USA) using primers pM_1380_f 5′-GACACATATGGGAGGAAATGTTGTGAGTGCCCCCGAGACG-3′ and pM_1380_r 5′-TTTTACTAGTTCAGGTGGCGCAGGGCG-3′ picked with primer-BLAST [22] and cloned in the pMINDKm^-^ plasmid [15], a modification of pMIND [24] lacking the kanamycin resistance gene, at the *Nde*I and *Spe*I restriction sites, to obtain the following plasmids: pMINDKm^-^:*msmeg_1380*, pMINDKm^-^:*msmeg_1380-19*, and pMINDKm^-^:*msmeg_1380-33*, containing, respectively, the *w.t. msmeg_1380* gene as well as its mutant variants from strains *atR19* and *atR33.* The resulting plasmids were electroporated in *M. smegmatis mc2 155* cells as described in [23].

*M. smegmatis* transformants were grown in Middlebrook 7H9 broth supplemented with hygromycin (50 μg/mL) and tetracycline (10 ng/mL) to midexponential phase (OD_600_ = 1.2). Afterwards the cultures were diluted in the proportion of 1:9:10 (culture:water:M290 medium) and 5 mL were poured as the top layer on Petri dishes with agarized M290 medium. Both top- and bottom-layers were supplemented with hygromycin (50 μg/mL) and tetracycline (10 ng/mL). The plates were allowed to dry for at least 30 min, afterwards sterile paper discs with impregnated imidazo[1,2-*b*][1,2,4,5]tetrazines were plated. The plates were incubated for 2–3 days at 37 °C, until the bacterial lawn was fully grown. Growth inhibition halos were measured to the nearest 1 mm. The experiments were carried out as triplicates; the average diameter and standard deviation (SD) were calculated.

### 4.5. Mycobacterial RNA Isolation and Real-Time qPCR

*M. smegmatis* strains were grown overnight in Middlebrook 7H9 broth to midexponential phase (OD_600_ = 1.0–1.2); cells from 10 mL culture were harvested by centrifugation for 10 min at 3000× *g* and washed by 1 mL of RNAprotect Bacteria Reagent (Qiagen, USA). Total RNA was extracted by homogenization in Trizol solution (Invitrogen, USA) [25], followed by phenol (pH = 4.5)-chloroform/isoamyl alcohol (25:24:1) purification and precipitation in high salt solution (0.8 M Na citrate, 1.2 M NaCl) with isopropanol. Remaining genomic DNA was removed by DNAse I, Amplification grade (Invitrogen, USA). 50 ng of total RNA was used for cDNA synthesis by iScript Select cDNA Synthesis Kit (Bio-Rad, USA). 1 ng of cDNA was used for real-time qPCR with the qPCRmix-HS SYBR kit (Evrogen, Russia) on a CFX96 Touch machine (Bio-Rad, USA). CFX Manager V 3.1 (Bio-Rad, USA) was used to analyze the qPCR results: relative normalized expression of three biological replicates was calculated as ΔΔCq and genes *sigA* and *ftsZ* were used as reference. The following primers were picked by primer-BLAST [22] for qPCR: q1380-f 5′-CTGCTCGACGAACCATGCGAAAC-3′ and q1380-r 5′-AAGGGTCTTGAGCCGAATCTCAACG-3′ (*MSMEG_1380*), q1382-f 5′-ACCACGCAGATCATGAACAACGACT-3′ and q1382-r 5′-GAAATCGTCGAAGTCCGCCAGATGA-3′ (*MSMEG_1382*), qsigAs-sm-f 5′-CGAGCTTGTTGATCACCTCGACCAT-3′ and qsigAs-sm-r 5′-CTCGACCTCATCCAGGAAGGCAAC-3′ (*sigA*), qftsZs-sm-f 5′-AGCAGCTCCTCGATGTCGTCCTT-3′ and qftsZs-sm-r 5′-GCCTGAAGGGCGTCGAGTTCAT-3′ (*ftsZ*).

## Figures and Tables

**Figure 1 pathogens-09-00166-f001:**
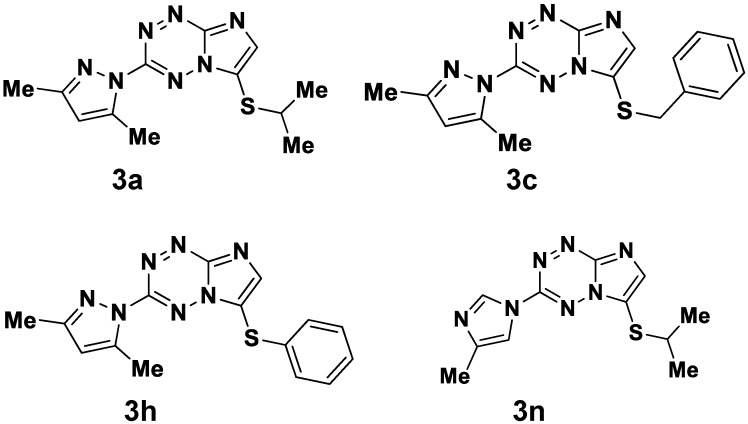
Chemical structures of imidazo[1,2-*b*][1,2,4,5]tetrazines [9].

**Figure 2 pathogens-09-00166-f002:**
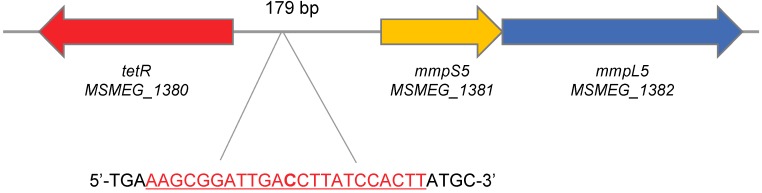
Schematic representation of the *mmpS5-mmpL5* operon structure in *M. smegmatis* genome. The putative operator sequence is shown in red.

**Figure 3 pathogens-09-00166-f003:**
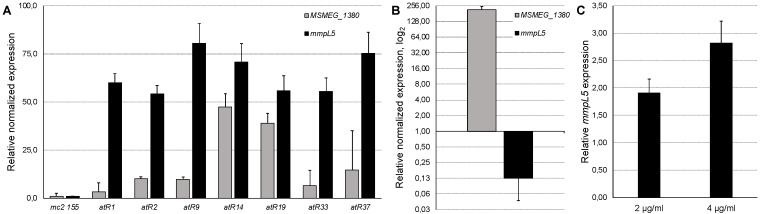
Relative expression levels of *mmpS5-mmpL5* operon genes in different conditions: expression levels of *MSMEG_1380* and *mmpL5* genes in spontaneous *M. smegmatis* imidazo[1,2-*b*][1,2,4,5]tetrazine-resistant mutants (**A**); expression levels of *MSMEG_1380* and *mmpL5* genes in *M. smegmatis* pMINDKm^-^:*msmeg_1380* (**B**); expression levels of the *mmpL5* gene after the addition of different concentrations of the compound **3a** (shown on the X-axis) (**C**). Error bars represent standard deviations from triplicates.

**Table 1 pathogens-09-00166-t001:** Bacterial strains used in the study.

Bacterial Strains		
Name	Comment	Origin
*M. smegmatis mc2 155*	Wild-type (*w.t.*) strain	
*M. smegmatis atR1*	Spontaneous mutant of *mc2 155.* Mutations: Y_52_H (TAC>CAC) in *MSMEG_1601*; del LLA_41-43_ (del GCTGCTCGC_480-488_) in *MSMEG_1380.*	[9]
*M. smegmatis atR2*	Spontaneous mutant of *mc2 155.* Mutations: Y_52_H (TAC>CAC) in *MSMEG_1601*; ins GC_425-426_ (frameshift) in *MSMEG_1380.*	[9]
*M. smegmatis atR9*	Spontaneous mutant of *mc2 155.* Mutations: Y_188_C (TAC>TGC) in *MSMEG_2087*; ins C_8_ (frameshift) in *MSMEG_1380.*	[9]
*M. smegmatis atR10*	Spontaneous mutant of *mc2 155.* Mutations: R_233_S (CGT>AGT) in *MSMEG_0641*; ins C_8_ (frameshift) in *MSMEG_1380.*	[9]
*M. smegmatis atR14*	Spontaneous mutant of *mc2 155.* Mutations: Y_52_H (TAC>CAC) in *MSMEG_1601*; ins G_448_ (frameshift) in *MSMEG_1380.*	[9]
*M. smegmatis atR19*	Spontaneous mutant of *mc2 155.* Mutations: Y_52_H (TAC>CAC) in *MSMEG_1601*; T_52_V (ACG>GTG) in *MSMEG_1380.*	[9]
*M. smegmatis atR33*	Spontaneous mutant of *mc2 155.* Mutations: ins VG_52-53_ (ins GTGGGC_154-159_) in *MSMEG_1380.*	[9]
*M. smegmatis atR37*	Spontaneous mutant of *mc2 155.* Mutation: del C 662 (frameshift) in *MSMEG_1380.*	[9]
*M. smegmatis atR1c*	Recombinant strain, mutation: del LLA_41-43_ (del GCTGCTCGC_480-488_) in *MSMEG_1380.*	This study
*M. smegmatis atR2c*	Recombinant strain, mutation: ins GC_425-426_ (frameshift) in *MSMEG_1380.*	This study
*M. smegmatis atR9c*	Recombinant strain, mutation: ins C_8_ (frameshift) in *MSMEG_1380.*	This study
*M. smegmatis atR14c*	Recombinant strain, mutation: ins G_448_ (frameshift) in *MSMEG_1380.*	This study
*M. smegmatis atR33c*	Recombinant strain, mutation: ins VG_52-53_ (ins GTGGGC_154-159_) in *MSMEG_1380*	This study
*M. smegmatis 0641c*	Recombinant strain, mutation: R_233_S (CGT>AGT) in *MSMEG_0641.*	This study
*M. smegmatis 1601c*	Recombinant strain, mutation: Y_52_H (TAC>CAC) in *MSMEG_1601*	This study
*M. smegmatis 2087c*	Recombinant strain, mutation: Y_188_C (TAC>TGC) in *MSMEG_2087*	This study

**Table 2 pathogens-09-00166-t002:** Imidazo[1,2-*b*][1,2,4,5]tetrazines MICs on *M. smegmatis* strains in liquid medium.

Compound	*M. smegmatis* Strains MICs, μg/mL								
	*mc2 155*	*atR1c*	*atR2c*	*atR9c*	*atR14c*	*atR33c*	*0641c*	*1601c*	*2087c*
**3a**	128	512	512	512	512	512	128	128	128
**3c**	64	>128 *	>128 *	>128 *	>128 *	>128 *	64	64	64
**3h**	128	>256 *	>256 *	>256 *	>256 *	>256 *	128	128	128
**3n**	64	256	256	256	256	256	64	64	64

* The compounds were not soluble at higher concentrations; bacterial growth was observed at the stated concentrations.

**Table 3 pathogens-09-00166-t003:** Growth inhibition halos, produced by imidazo[1,2-*b*][1,2,4,5]tetrazines on *M. smegmatis* strains.

Compound	Concentration, nmole/disc	Growth Inhibition Halo, mm			
		*M. smegmatis* Transformants			
		pMINDKm^-^	pMINDKm^-^:*msmeg_1380*	pMINDKm^-^:*msmeg_1380-19*	pMINDKm^-^:*msmeg_1380-33*
**3a**	300	9.8 ± 1.5	17.0 ± 3.6	8.8 ± 0.8	8.2 ± 0.6
**3c**	300	7.0 ± 0.8	15.0 ± 2.9	6.3 ± 0.5	6.3 ± 0.5
**3h**	40	6.7 ± 0.5	11.5 ± 0.4	6.3 ± 0.5	6.5 ± 0.4
**3n**	100	9.7 ± 2.4	16.0 ± 0.8	9.2 ± 1.3	9.3 ± 1.2

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
