# Peer review of "MmpS5-MmpL5 Transporters Provide Mycobacterium smegmatis Resistance to imidazo[1,2-b][1,2,4,5]tetrazines"

_pathogens, 2020, doi:10.3390/pathogens9030166_

Round 1
Reviewer 1 Report
The manuscript by Maslov et al describes the characterization of several M. smegmatis mutants with expected resistance to novel and potential anti-tuberculosis compounds. The authors described the creation of mutants in 4 different M. smegmatis genes with further MIC testing against the compound. Based on obtained MICs as well as gene expression analysis, authors determine that resistance to imidazole[1,2-b][1,2,4,5]tetrazines is most likely due to mutations in MSMEG_1380. Authors hypothesized that MSMEG_1380 is a repressor of MmpS5-MmpL5 due to observed increased expression in the mutants as well as repression when MSMEG_1380 is overexpressed.
Overall the manuscript is well written and scientifically sound. Nonetheless it could be improved by:
- Demonstrate low (or null) expression of MSMEG_1380 in the mutants. Perhaps some mutants have similar expression as w.t. but the protein structure is altered. Regardless, measuring the expression of MSMEG_1380 in the different mutants may provide additional information on the role of each mutation.
- Add results for w.t. in Figure 2A.
Author Response
Thank you for a constructive review of our manuscript. Please find below a point-by-point response to your comments:
“1) Demonstrate low (or null) expression of MSMEG_1380 in the mutants. Perhaps some mutants have similar expression as w.t. but the protein structure is altered. Regardless, measuring the expression of MSMEG_1380 in the different mutants may provide additional information on the role of each mutation.”
Response: We have now included the results of the expression levels of MSMEG_1380 in Figure 3A, together with expression levels of mmpL5. The mutations within MSMEG_1380 apparently lead to its repressor function loss, and, as we observe here, the mmpS5-mmpL5 has a typical regulatory organization, when both the operon and the repressor are dependent on one promotor and operator (the repressor is self-regulatory in this case), thus we observe increased expression of both the MSMEG_1380 and mmpL5 genes in the mutant strains. This observation is now discussed briefly in lines 183-213. The caption of Figure 3 was edited (line 218) in order to correspond to the updated figure.
“2) Add results for w.t. in Figure 2A.”
Response: We have now added the expression levels of MSMEG_1380 and mmpL5 genes for the w.t. strain (mc2 155) on Figure 3A.
Reviewer 2 Report
In this manuscript, the authors reported the role of the mutations in mycobacterial drug resistance to four imidazo[1,2-b][1,2,3,4]tetrazine derivatives using approaches such as the construction of targeted mutants, overexpression of the mutant and wild-type genes, and gene-expression studies. Tuberculosis being one of the top ten causes of human deaths worldwide, requires considerable attention from the scientific community. So, the current work demonstrated here by authors is timely. The experiments conducted here to demonstrate the role of MSMEG_1380 in M. smegmatis resistance to imidazo[1,2-b][1,2,4,5]tetrazines are logical and well described. Thus, I recommend that this manuscript be accepted for publication in Pathogens after minor revisions. Below are some comments for the authors to consider.
- In the first paragraph of the introduction, the authors provided a very brief discussion about TB. Please provide more information and pitfalls of the current treatment options for TB treatment, to highlight the importance of developing novel therapies that require shorter treatment times (E.g. It is estimated that about one-quarter of the world's population has latent TB. The major obstacle is antibiotic resistance; due to the lack of compliance from patients during a long (over six months) and a complex course of treatment).
- Please provide a figure with chemical structures of compounds 3a, 3c, 3h, and 3n from reference 7 (Maslov et al. European Journal of Medicinal Chemistry 178 (2019) 39-47). These four compounds are extensively discussed in the manuscript, so it is appropriate to provide their chemical structures.
- There are several long sentences in the manuscript that are hard to read; please consider splitting them into multiple sentences (E.g., line:37, line 138, etc.,)
- Line 35: Please consider rewriting the sentence.
- Line 41: binding pocket…..
- Line 77: as well as two of its mutant variants…..
- Please carefully proof-read the manuscript to eliminate grammatical errors.
Author Response
Thank you for the constructive review of our manuscript. Please find below the point-by-point response to your comments.
“1) In the first paragraph of the introduction, the authors provided a very brief discussion about TB. Please provide more information and pitfalls of the current treatment options for TB treatment, to highlight the importance of developing novel therapies that require shorter treatment times (E.g. It is estimated that about one-quarter of the world's population has latent TB. The major obstacle is antibiotic resistance; due to the lack of compliance from patients during a long (over six months) and a complex course of treatment).”
Response: We have now expanded the introduction, adding the WHO estimates on the global number of people infected by M. tuberculosis (lines 31-32), as well the need of novel regimens, that might shorten the anti-TB therapy times, citing Muñoz-Torrico et al. (lines 35-38).
“2) Please provide a figure with chemical structures of compounds 3a, 3c, 3h, and 3n from reference 7 (Maslov et al. European Journal of Medicinal Chemistry 178 (2019) 39-47). These four compounds are extensively discussed in the manuscript, so it is appropriate to provide their chemical structures.”
Response: We have now added Figure 1 (lines 72-73) with a corresponding capture, being mentioned in the text on line 61.
“3) There are several long sentences in the manuscript that are hard to read; please consider splitting them into multiple sentences (E.g., line:37, line 138, etc.,)”
Response: We have now split the sentence on line 37 into 3 sentences (lines 41-59), adding some minor stylistic changes. The long sentence on line 138 was also rewritten, and split in two sentences: the first one describing the primary biological role of the MmpS5-MmpL5 system, and the second one focusing on its role in drug resistance (lines 245-249). The following sentence (lines 249-334) has also been slightly edited in order to be clearer.
“4) Line 35: Please consider rewriting the sentence.”
Response: We have now changed the structure of the sentence. Now it first states the role of ESTPKs, and after that makes a conclusion, that they are attractive targets for drug development: “Eukaryotic-type serine-threonine protein-kinases (ESTPKs) play a key role in M. tuberculosis life cycle regulation, controlling some of its vital aspects such as cell division and survival within host macrophages, and thus represent attractive targets for drug development [7,8].” (lines 39-41).
“5) Line 41: binding pocket
6) Line 77: as well as two of its mutant variants”
Response: Thank you for these recommendations, we made the recommended changes (lines 57 and 153).
“7) Please carefully proof-read the manuscript to eliminate grammatical errors.”
Response: We did the proofreading and made some minor edits, including changing “w.t.” to italic throughout the text; line 35 – “global threat to world-wide TB control”; line 152 changed from “To further investigate” to “In order to investigate further”; line 156 – added the article “the” before “overexpression”; line 235 – “lead” changed to “led”; line 236 “we have observed” instead of “we’ve observed”.